# Rainfall Map from Attenuation Data Fusion of Satellite Broadcast and Commercial Microwave Links

**DOI:** 10.3390/s22187019

**Published:** 2022-09-16

**Authors:** Fabio Saggese, Vincenzo Lottici, Filippo Giannetti

**Affiliations:** 1Department of Electronic System, Aalborg University, 9220 Aalborg, Denmark; 2Department of Information Engineering, University of Pisa, 56122 Pisa, Italy

**Keywords:** rainfall rate estimation, rainfall map estimation, microwave propagation, rain attenuation, earth–satellite link, commercial microwave link

## Abstract

The demand for accurate rainfall rate maps is growing ever more. This paper proposes a novel algorithm to estimate the rainfall rate map from the attenuation measurements coming from both broadcast satellite links (BSLs) and commercial microwave links (CMLs). The approach we pursue is based on an iterative procedure which extends the well-known GMZ algorithm to fuse the attenuation data coming from different links in a three-dimensional scenario, while also accounting for the virga phenomenon as a rain vertical attenuation model. We experimentally prove the convergence of the procedures, showing how the estimation error decreases for every iteration. The numerical results show that adding the BSL links to a pre-existent CML network boosts the accuracy performance of the estimated rainfall map, improving up to 50% the correlation metrics. Moreover, our algorithm is shown to be robust to errors concerning the virga parametrization, proving the possibility of obtaining good estimation performance without the need for precise and real-time estimation of the virga parameters.

## 1. Introduction

In recent years, the higher and higher occurrence of extreme phenomena related to climate change has considerably spurred the demand of accurate and real-time rainfall maps. To this end, classical methods for rainfall measurement, i.e., surface sensors, weather radars and satellite systems have been intensively used. These solutions, however, require non-negligible installation and operating costs, while the measurements they offer have a temporal and spatial resolution that is often not sufficient for the tasks of interest.

In the last decades, the use of opportunistic sensors for rainfall estimation has opened the way to various methods based on signal processing techniques applied to attenuation measurements of existing commercial microwave links (CMLs). Several different strategies have been published on this topic such as [1,2], and in the sequel, [3,4,5]. As an alternative option, it has also been proposed to obtain the rain attenuation contribution from the overall attenuation measured at the CML receiver [6,7,8]. In addition, exploiting the measurements of the attenuation data available at the CML receiver, the generation of rainfall maps through the inverse distance weighting (IDW) algorithm or tomographic estimation has been addressed in [9,10,11] and in [1,12,13], respectively.

Recently, a significant research effort has been devoted also to the estimation of the rainfall rate from the received signal level at the ground station of direct-to-home (DTH) broadcast satellite links (BSLs) [14,15,16,17,18,19,20]. Hence, taking advantage of the limited cost and ease of installation of the commercial-grade BSL receivers for DTH broadcast, the rationale of our current work consists of effectively merging together the attenuation data coming from both the CML and BSL receivers. Differently from our preliminary approach  [21], (to the best of our knowledge) the first rainfall estimation system based on fusing together such measurements, here, we propose a modified GMZ algorithm [9] which properly handles the BSL and CML attenuation data over a three-dimensional (3-D) scenario of interest, where the vertical variations of rain intensity affecting the satellite links are considered.

The main contributions are as follows.

We consider the *virga* phenomenon, i.e., a variation of rainfall rate with respect to the height due to a gradient of environmental parameters such as humidity, which may cause evaporation or sublimation of rain [22];Based on the virga model, we propose a hybrid 3-D modified version of the GMZ algorithm [9,11] which provides the rainfall rate estimation by properly merging together the attenuation measurements collected at the BSL and CML receivers;We numerically show that the root mean square estimation error steadily decreases at each step of the iterative procedure, thus leading to a stable solution for the rainfall intensity although the underlying optimization problem is not convex;We prove that a few BSLs placed in locations scarcely covered by CMLs can act as gap-fillers, with the result of notably improving the rainfall estimation performance with respect to conventional schemes based on either CMLs or BSLs only;The robustness of the proposed approach is achieved even in case of non-ideal knowledge of the parameters describing the virga phenomenon, which is significant indication that the hard task of its real-time characterization [22,23] can be avoided.

The organization of the paper is as follows. In Section 2, the environmental scenario is described, i.e., we define the geometrical position of CMLs and BSLs in Section 2.1, the rainfall model for both horizontal and vertical planes in Section 2.2, and the quantization procedure to obtain the rainfall map in Section 2.3. In Section 3, we outline the hybrid iterative optimization algorithm, in Section 4, we discuss the numerical results, and finally, some conclusions are drawn in Section 5.

## 2. Environmental Scenario

### 2.1. Geometrical Model

The area of interest has a parallelepiped shape P with a square base of area *B* and height equal to the rain height $h_{0}$, defined as in [24] and assumed to be constant over *B*. The coordinates (measured in km) of each point inside P are referenced as $p={\left[x,y,z\right]}^{T}\in P$, ${\left[\;\cdot \;\right]}^{T}$ denoting the transpose of a vector, with
(1)$$\begin{array}{cc} -\sqrt{B}/2 & \leq x\leq \sqrt{B}/2,\\ -\sqrt{B}/2 & \leq y\leq \sqrt{B}/2,\\ 0 & \leq z\leq h_{0}. \end{array}$$

The above-defined geographical scenario to be monitored includes $N_{\mathrm{CML}}$ horizontal CMLs along with $N_{\mathrm{BSL}}$ slanted BSLs, whose available data measurements are properly gathered up and fused together; consider Figure 1 and Figure 2 as examples. After indicating with *n* the link index, either terrestrial or satellite, and denoting the overall number of links as $N=N_{\mathrm{CML}}+N_{\mathrm{BSL}}$, the *n*-th link is geometrically delimited by $p_{n,1}={\left[x_{n,1},y_{n,1},z_{n,1}\right]}^{T}\in P$ and $p_{n,2}={\left[x_{n,2},y_{n,2},z_{n,2}\right]}^{T}\in P$, 1≤n≤N.

Hence, we obtain that: (*i*) For CMLs, $z_{n,1}=z_{n,2}$; (*ii*) For BSLs, $z_{n,2}=h_{0}$; (*iii*) The length of the *n*-th link results as
(2)$$L_{n}=\sqrt{{\left(p_{n,1}-p_{n,2}\right)}^{T}\left(p_{n,1}-p_{n,2}\right)},$$
while its angle of elevation over the horizontal plane x−y is given by
(3)$$\theta_{n}=\arctan2\left(z_{n,2}-z_{n,1},x_{n,2}-x_{n,1}\right).$$

Exploiting the former equations, it can be further remarked that: (*iv*) $\theta_{n}=0$ for all the CMLs; (*v*) The BSLs are characterized by a slanted path between the satellite transmitter and the ground receiver; (*vi*) Due to the distance of the geostationary satellite from the scenario and assuming that $\sqrt{B}$ is around few kilometers, $\theta_{n}=\theta$, for all the BSLs, i.e., the elevation angle can be assumed constant for all the links pointing to the same satellite. (The case of using BSLs from different satellites visible at different azimuth and elevation angles is under study.)

### 2.2. Rainfall Model

The specific rain attenuation, in dB/km, experienced at a location p lying along the *n*-th wireless communication link with carrier frequency $f_{n}$ in the 10–30 GHz frequency range depends (with a good approximation) on the local rainfall rate r(p), in mm/h, according to the *power law formula* [25]
(4)$$\alpha_{n}\left(p\right)=a_{n}{\left[r\left(p\right)\right]}^{b_{n}}\;\left[\mathrm{dB}/\mathrm{km}\right],$$
where $a_{n}$ and $b_{n}$ are empirical coefficients (assumed to be known throughout the paper) relying on $f_{n}$ and on the polarization, i.e., horizontal, vertical or circular, [14,26,27]. It is also worth emphasizing that the rain attenuation represents only one of the contributions to the total attenuation affecting the *n*-th link. State-of-the-art works, however, have shown that the rainfall contribution can be reliably extracted from the measurements of the total attenuation [7,14,15]. Furthermore, for the sake of simplicity, we assume the ITU model based on stratiform rain with two layers only, i.e, *solid* and *liquid* [24] (A more accurate model considering a third layer, named *melting* layer between the solid and the liquid layers, was proposed and investigated in [14,16].), as shown in Figure 1, where, for instance, the link 1 is a CML having length $L_{1}$, attenuation $A_{1}$ and elevation angle $\theta_{1}=0$, and the link 2 is a BSL with wet path length, i.e., the portion of the path inside the liquid layer, $L_{2}$, attenuation $A_{2}$ and elevation angle $\theta_{2}$. It can be remarked that the solid layer marginally influences the attenuation, thus explaining the reason why the BSL link length is defined as the wet-path only.

We model the rainfall rate as the spatially-continuous random process, expressed in mm/h, r(p)∈P, i.e., with values depending on the position within the 3-D scenario. Hence, according to (Equation 4), the total rain attenuation, in dB, experienced by the *n*-th link reads as
(5)$$A_{n}=\int_{p_{n,1}}^{p_{n,2}}a_{n}{\left[r\left(p\right)\right]}^{b_{n}}dp\;\left[\mathrm{dB}\right].$$
As detailed hereafter, the rainfall will be modeled differently in a generic x−y horizontal (H) plane at height *z*, denoted as $\pi_{H}\left(z\right)$, and in a generic x−z vertical (V) plane at coordinate *y*, denoted as $\pi_{V}\left(y\right)$.

#### 2.2.1. The $\pi_{H}\left(z\right)$ Plane

Let us consider *K* points lying on the same H-plane at height *z*, collect them into the set K, i.e., $p_{k}={\left[x_{k},y_{k},z\right]}^{T}\in K$, and assume that the relevant rainfall rates $r(p_{k})$, 1≤k≤K, are known. Then, let us consider $p_{u}={\left[x_{u},y_{u},z\right]}^{T}$, which lies on the same plane $\pi_{H}\left(z\right)$, but with an unknown rainfall rate. The rainfall rate at $p_{u}$ can be estimated according to the Shepard’s inverse distance weighting (IDW) method [10,28,29]
(6)$$r\left(p_{u}\right)=\frac{\sum_{k=1}^{K}W_{u,k}\;r\left(p_{k}\right)}{\sum_{k=1}^{K}W_{u,k}},\;\left[\mathrm{mm}/h\right].$$

The adimensional weights $W_{u,k}\in \left[0,1\right]$ in (Equation 6) are expressed as
(7)$$W_{u,k}={\left[\frac{{\left(1-d_{u,k}/\Gamma \right)}^{2}}{{\left(d_{u,k}/\Gamma \right)}^{2}}\right]}^{+},$$
where ${\left[\;\cdot \;\right]}^{+}=\max\left\{0,\;\cdot \;\right\}$, the constant Γ is the *radius of influence*, i.e., the radius of a circumference centered on $p_{u}$ and lying on $\pi_{H}\left(z\right)$, which is suitably set (In [9], the radius of influence depends on the density of the data points and is adaptively chosen so as to include at least five data points.) to encompass those points on the H plane whose rainfall rates are assumed to appreciably contribute to the evaluation of the rainfall rate at $p_{u}$, and
(8)$$d_{u,k}=\sqrt{{\left(p_{u}-p_{k}\right)}^{T}\left(p_{u}-p_{k}\right)}$$
denotes the distance between the point with unknown rainfall rate and the *k*-th point of the set K. Consequently, the rainfall rate in (Equation 6) is evaluated considering only those points for which $d_{u,k}<\Gamma$.

#### 2.2.2. The $\pi_{V}\left(z\right)$ Plane

Let us take into consideration the following two points, both lying on the same vertical line, i.e., $p_{k}={\left[x,y,z_{k}\right]}^{T}$, where the relevant rainfall rate $r(p_{k})$ is known, and $p_{u}={\left[x,y,z_{u}\right]}^{T}$, where the relevant rainfall rate is instead unknown. Additionally, we consider the presence of a vertical gradient of the rainfall rate, the so-called *virga phenomenon*. An accurate yet analytically involved expression of the upwards vertical variation of the rainfall rate is provided by [30]. Such analytical model, however, involves many parameters, which vary according to the type of the precipitation, and so, in general, are hard to estimate [22]. Nevertheless, experimental results for temperate climates presented in [23] show that the dependence of the rainfall rate with the height can be accurately modeled by a simple linear law, i.e.,
(9)$$\nu \left(p_{u},p_{k}\right)={\left[r\left(p_{k}\right)-g\left(x,y\right)\left(z_{u}-z_{k}\right)\right]}^{+},\;\left[\mathrm{mm}/h\right],$$
where $\nu (p_{u},p_{k})$ is the rainfall rate evaluated at $p_{u}$ by applying the linear model from the knowledge of the rainfall rate at $p_{k}$. By convention, the gradient of the rainfall rate g(x,y), expressed in mm/h/km, is assumed to be positive if the rainfall rate increases with the altitude, so that the highest value is attained at height $h_{0}$. Further, if the model (Equation 9) yields a null rain rate at a given height, the rain rates will be zero for all the points below, down to the ground. For the sake of simplicity, we assume a constant gradient all over the scenario, i.e., g(x,y)=g,∀x,y such that $p={\left[x,y,z\right]}^{T}\in P$. (The proposed procedure can be easily generalized also to the case of a non-uniform gradient over the area of interest).

#### 2.2.3. Overall Model

Merging together the assumptions on the planes $\pi_{H}\left(z\right)$ and $\pi_{V}\left(y\right)$, along with assuming that the rainfall rates $r(p_{k})$ of the set of *K* points $p_{k}$ are known, the rainfall rate at the generic point $p_{u}\in P$ can be estimated as
(10)$$r\left(p_{u}\right)=\frac{\sum_{k=1}^{K}W_{u,k}\;\nu \left(p_{u},p_{k}\right)}{\sum_{k=1}^{K}W_{u,k}},\;\left[\mathrm{mm}/h\right].$$

### 2.3. Quantization

The distribution of the rainfall rate at the base of the volume P, i.e., at z=0, is obtained by building a spatially quantized two-dimensional (2-D) map of J×J pixels, each pixel with an area of $\Delta =B/J^{2}$. The center of each pixel is denoted with $c_{j}$, $1\leq j\leq J^{2}$, while the *grid points* consists of the overall set $C={\left\{c_{j}\right\}}_{j=1}^{J^{2}}$. Hence, assuming that the rainfall rate does not significantly change over Δ, $r(c_{j})$ stands for the rainfall rate for each pixel. Therefore, the overall rainfall map the algorithm yields is given by $r\left(C\right)={\left[r\left(c_{1}\right),\ldots ,r\left(c_{J^{2}}\right)\right]}^{T}$.

Following the approach of [11], the quantization process is applied to the links as well. We subdivide each link into segments with length *D* where the rainfall rate can be assumed to be nearly constant, thus obtaining for the *n*-th link a number of $Q_{n}$ intervals equal to
(11)$$Q_{n}=\left\lceil L_{n}\cos\theta_{n}/D\right\rceil ,\;1\leq n\leq N,$$
where $L_{n}\cos\left(\theta_{n}\right)$ is the projection of the BSL or CML link onto the base, and $\left\lceil \;\cdot \;\right\rceil$ takes the nearest lower integer of the argument. The center of the *q*-th segment, $1\leq q\leq Q_{n}$, for the *n*-th link, 1≤n≤N, is called *data point*, with coordinates $d_{n,q}={\left[x_{n,q},y_{n,q},z_{n,q}\right]}^{T}$. All the data points of the *n*-th link are then collected in the set $Q_{n}={\left\{d_{n,j}\right\}}_{j=1}^{Q_{n}}$, being the corresponding rainfall rates denoted as
(12)$$r\left(Q_{n}\right)={\left[r\left(d_{n,1}\right),\cdots ,r\left(d_{n,Q_{n}}\right)\right]}^{T},\;\left[\mathrm{mm}/h\right].$$

## 3. Estimation Algorithm

An iterative estimation algorithm of the rainfall rate for the data points $r(Q_{n})$ and grid points r(C) within the scenario of interest is outlined by combining the IDW estimation algorithm in (Equation 10) with a proper constrained optimization problem (OP).

As a first step, let us consider the estimation of the rainfall rate of the data points of the *n*-th link for the *i*-th iteration. We employ the rainfall rate corresponding to the data points not belonging to the *n*-th link obtained at iteration i−1, i.e., $r{\left(Q_{\ell }\right)}^{\left(i-1\right)}$, 1≤ℓ≤N, ℓ≠n, to estimate the rainfall rate of the *n*-th link through (Equation 10), denoted as $\hat{r}\left(Q_{n}\right)$, 1≤n≤N. Specifically, the distance between the desired $r(Q_{n})$ and the one estimated from the other links $\hat{r}\left(Q_{n}\right)$ is minimized, under the constraint that the overall rainfall attenuation $A_{n}$ measured over the *n*-th link be
(13)$$A_{n}=\sum_{q=1}^{Q_{n}}\frac{L_{n}\cos\theta_{n}}{Q_{n}}\;a_{n}{\left[r\left(d_{n,q}\right)\right]}^{b_{n}},$$
having approximated the integral (Equation 5) as the summation over all the intervals the *n*-th link has been subdivided into, whose centers are located at the corresponding data points. Hence, the OP can be formalized as
(14)$$\begin{array}{cc} \; & \underset{r(Q_{n})}{arg min}||r\left(Q_{n}\right)-\hat{r}\left(Q_{n}\right){\left|\right|}^{2}\\ s.t. & \;\frac{A_{n}Q_{n}}{a_{n}L_{n}\cos\theta_{n}}-\sum_{q=1}^{Q_{n}}{\left[r\left(d_{n,q}\right)\right]}^{b_{n}}=0, \end{array}$$
as a 3-D generalization of the OP described in [9]. Since the constraint is not affine, the OP (Equation 14) is not convex, and thus, we argue that the optimal solution is not unique. Our approach is to apply the gradient descent-based method to converge to (at least) a local optimal solution [31], i.e., $r{\left(Q_{n}\right)}^{\left(i\right)}$.

As a second step, the whole optimal set of data points is obtained, running the OP (Equation 14) for each link *n*, until convergence is reached, or equivalently, when $e_{i}<\epsilon$, with an arbitrary ϵ>0, being $e_{i}$ the error of the optimization procedure at iteration *i*-th, i=1,2,…, defined as
(15)$$e_{i}=\sqrt{\sum_{n=1}^{N}||r{\left(Q_{n}\right)}^{\left(i\right)}-r{\left(Q_{n}\right)}^{\left(i-1\right)}{\left|\right|}^{2}}.$$

As a third and final step, using the rainfall rate at the data points as input, the rainfall rate estimation on the grid points is performed using (Equation 10), thus obtaining the map r(C).

The proposed iterative algorithm is outlined in Algorithm 1. The rainfall rates are initialized with the values $r{\left(Q_{n}\right)}^{\left(0\right)}=R_{n}$, 1≤n≤N, where
(16)$$R_{n}={\left(\frac{A_{n}}{a_{n}L_{n}}\right)}^{\frac{1}{b_{n}}}\;\left[\mathrm{mm}/h\right].$$
is obtained by inverting (Equation 4) and assuming
(17)$$\alpha_{n}^{\left(0\right)}=\frac{A_{n}}{L_{n}}\;\left[\mathrm{dB}/\mathrm{km}\right]$$
as an initial guess of the specific attenuation $\alpha_{n}\left(p\right)$ for all the data points of the *n*-th link.
**Algorithm 1:**Iterative OP.
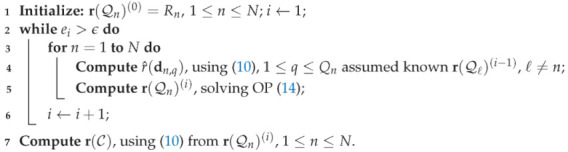


## 4. Numerical Results

To corroborate the effectiveness of the proposed approach, a set of simulations were run for the scenario of interest. The simulation makes use of a synthesized rain profile, and the network topology shown in Figure 2, whose parameters are presented in Table 1. (The evaluation of the performance using experimental data is under investigation.) We simulate a square-shaped scenario with side length $\sqrt{B}$, which contains a given number of CMLs and BSLs at known locations. For the sake of simplicity, we assume that all the links are operating with the same carrier frequency $f_{n}$, and the same polarization. (We assume frequency values typically employed by CMLs [9] to be used also for the BSLs, for simplicity. While BSLs’ carriers are usually in the Ku band (i.e., 10–13 Hz) [16], we remark that the effect of the operational frequency is completely described by the parameters $a_{n}$ and $b_{n}$ [27]; therefore, we may use the same frequency for both the set of links without loss of generalization.) All the (horizontal) CMLs have 0° elevation angle. Moreover, we also assume that all the BSLs are pointed toward the same the satellite, laying on the local meridian, so that, in a city located within 43° and 44° parallels (e.g., Pisa, Italy), the corresponding elevation angle result is approx. $\theta_{n}=39.5$°; this value is set for any BSL terminal. Based on (Equation 4), the coefficients $a_{n},b_{n}$, computed according to [27], are used to model the rain attenuation in the whole scenario, as in [14,26]. The value of the radius of influence Γ is computed in order to have non-zero weight for at least *five* other data points in the evaluation of Equation (Equation 7), according to [9]. The simulated rainfall intensity in mm/h is synthesized as a 2-D Gaussian-shaped spatial distribution on the x−y plane with standard deviation $\sigma_{G}$, and peak value *R* mm/h located at $p_{G}={\left[x_{G},y_{G},0\right]}^{T}$, as in [10].Moreover, the simulated precipitation is assumed to experience a fixed vertical gradient $g_{G}$, in mm/h/km, on the x−z plane, to take into account for the virga phenomenon. The resulted precipitation rainfall rates are generated at the same position of the grid points of the output estimated map C (see Section 2.3), and it is denoted as $\bar{r}\left(C\right)={\left[\bar{r}\left(c_{1}\right),\cdots ,\bar{r}\left(c_{J^{2}}\right)\right]}^{T}$.To obtain the quantized data points, the length *D* where the rainfall rate can be assumed constant (see Section 2.3) is set lower than 1/10 of the diameter of the rainfall phenomenon, as suggested in [9].

The overall accuracy performance of the estimation algorithm is quantified by both the root mean square error (RMSE) $\varepsilon_{\mathrm{RMS}}$
(18)$$\varepsilon_{\mathrm{RMS}}=\sqrt{\frac{{\left|\left|\bar{r}\left(C\right)-r\left(C\right)\right|\right|}^{2}}{J^{2}}}\;\left[\mathrm{mm}/h\right],$$
and the (adimensional) correlation coefficient ρ
(19)$$\rho =\frac{\mathrm{cov}\{\bar{r}\left(C\right),r\left(C\right)\}}{\mathrm{std}\left\{\bar{r}\left(C\right)\right\}\mathrm{std}\left\{r\left(C\right)\right\}}$$
where cov{·} and std{·} denote the covariance and the standard deviation operators, respectively. The accuracy contribution provided by the slanted satellite links in the estimate of the rain intensity map is affected by how the degree of accuracy of the gradient model. It is worth recalling that accurate real-time measurements of the vertical rain gradient are difficult to achieve as they would require costly and complex instrumentation; see, e.g., [22]. To evaluate the performance of the algorithm under realistic conditions, the gradient $g_{\mathrm{alg}}$ is adopted, which is not necessarily equal to the *true* value $g_{G}$ used in the generation of the synthetic rain. Numerical results are presented, indeed, under both the assumptions $g_{\mathrm{alg}}=g_{G}$, i.e., ideal knowledge of the current virga phenomenon, and $g_{\mathrm{alg}}\neq g_{G}$, i.e., imperfect knowledge.

*Convergence behavior of the algorithm*. To quantify the convergence behavior through an analytical approach was found to be too complex, and therefore, we resort to simulations. The error $e_{i}$ (Equation 15) is averaged for 1000 simulation runs. The resulting average error ${\bar{e}}_{i}$ is then plotted in Figure 3 for the first 100 iterations. As becomes apparent, the average error steadily decreases, thus providing the experimental evidence about the convergence of the proposed procedure.

*Rainfall rate estimation for a single realization*. A single realization of the rainfall rate is estimated for the topology depicted in Figure 2, in the presence of a given 2-D Gaussian rain intensity profile with peak value R=15 mm/h. Figure 4a shows the $\pi_{H}\left(0\right)$ horizontal plane of the scenario at ground level, including the following links: (*i*) A mesh of 21 CMLs (as white lines); (*ii*) The ground projections of 8 BSLs (as black lines). The rain cell is also visible at the top left of the map. The black circumference, which is centered on the peak of the precipitation and has radius $\sigma_{G}$, is shown for reference as a core area of the precipitation. Figure 4b shows the $\pi_{V}\left(y_{G}\right)$ vertical plane of the scenario, where the CMLs appear as near-ground horizontal white lines, while the BSLs are the slanted black lines reaching $h_{0}$. Also depicted is the vertical profile of the synthetic precipitation, characterized by a virga effect with gradient $g_{R}=5$ mm/h/km. Figure 4c,d offers the precipitation maps generated by the proposed algorithm, assuming a perfectly estimated vertical gradient, i.e., $g_{\mathrm{alg}}=g_{G}$. In both cases, the rainfall intensity distribution and the number ($N_{\mathrm{CML}}=21$) along with the geometry of the CMLs are the same. In case (c), there are no BSLs; thus, the estimation procedure relies on the pure GMZ algorithm [9,10], whose performance results in $\varepsilon_{\mathrm{RMS}}=5.715$ and ρ=0.470. In case (d), there are eight BSLs supplementing the measurements of the CMLs. Employing the proposed algorithm, the performance improves to $\varepsilon_{\mathrm{RMS}}=1.981$, i.e., −34%, and ρ=0.934, i.e., +50%. Hence, fusing BSL data through our approach leads to better estimation performance, proving the possibility of using, either already installed or purposely installed, satellite receivers as gap-fillers in CML networks.

*Rainfall rate estimation average performance*. To quantify the average performance, 1000 different scenarios are considered, each of them with randomly positioned BSLs, while the CMLs are randomly selected from the network topology shown in Figure 2. For every scenario, 50 different random positions of the rain cell $p_{G}$ are generated. Figure 5 shows the near-ground overall performance in terms of $\varepsilon_{\mathrm{RMS}}$ (Figure 5a) and ρ (Figure 5b) as a function of $N_{\mathrm{BSL}}$. The results are for different values of $N_{\mathrm{CML}}$, while keeping fixed R=15 mm/h and $g_{\mathrm{alg}}=g_{G}$. Again, when $N_{\mathrm{BSL}}=0$, the algorithm coincides with the pure GMZ algorithm. On the other hand, we present also the results for $N_{\mathrm{CML}}=0$ to show the performance obtained by collecting measures from pure BSL approaches, such as [14,15,16]. For both RMSE and ρ metrics, the boost in performance given by adding BSLs is greater than the one obtained by adding CML, as long as a minimum number of CML is present in the scenario. For example, for $N_{\mathrm{CML}}=7$, we can halve the RMSE by increasing the number of BSLs from 8 to 13 (i.e., inserting five BSL terminals in the scenario). To obtain a similar boost in performance with the CML, we need to triple the number of microwave links in the network. On the contrary, when no CMLs are present in the network, the estimation procedure is not able to reach good estimation performance. This phenomenon is partially due to the limited length of the projection of the BSL on the x−y plane. In fact, to cover the whole scenario, a large number of satellite terminals are needed. Another reason for this drop in performance may be due to the (approximate) parallel links obtained by pointing all terminals toward the same satellite, which could generate errors in the estimation of the data points’ rain rate. Nevertheless, the quantitative impact of this phenomenon on estimation performance is still under study. The results further support the choice of CML and BSL data fusion for improving the estimation performance.

Figure 6 illustrates ρ as a function of $g_{\mathrm{alg}}$ for $N_{\mathrm{CML}}=21$, a different number of $N_{\mathrm{BSL}}$, and with: (a) $g_{G}=0$, (b) $g_{G}=3$, (c) $g_{G}=6$ mm/h/km. The plots emphasize that the performance is slightly influenced by the error of the gradient, which gives minimal effects in the case $g_{G}=3$ mm/h/km. This proves that the proposed rainfall estimation technique does not require a precise real-time assessment of the rain gradient. In fact, just the use of a suitably selected constant value of the rain gradient (e.g., the average or a typical value) is enough to guarantee a small estimation error of the rainfall intensity.

## 5. Conclusions

In this paper, a novel hybrid procedure was illustrated to obtain the rainfall rate map estimation from fusing together the attenuation data collected at the receivers of both commercial microwave and broadcast satellite links. The proposed algorithm consists of the modified version of the GMZ one which has been properly extended to: (*i*) Apply the estimation procedure to a three-dimensional scenario; (*ii*) The virga phenomenon experienced by the BSLs; (*iii*) Merge the attenuation data from the mixed available CMLs and BSLs; (*iv*) Take into account the experimentally proven convergence of the iterative estimation algorithm. The numerical results show the boost in the accuracy performance provided by employing BSL terminals together with CMLs. Moreover, we show that the proposed algorithm is robust to possible errors in the parameters used to model the virga effect, reducing the need for precise real-time estimation of those parameters. Combined with the low installation cost of BSL terminals, the proposed algorithm ensures that the BSLs can act as as gap-fillers of an pre-existing CML network.

## Figures and Tables

**Figure 1 sensors-22-07019-f001:**
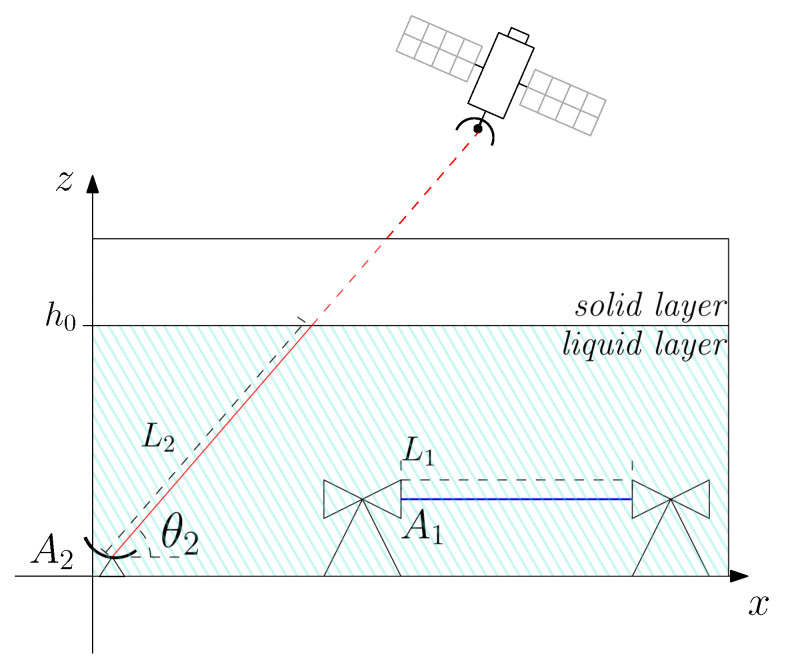
Stratiform rain model and geometry of the scenario. Link 1: CML, with length $L_{1}$, attenuation $A_{1}$ and elevation angle $\theta_{1}=0$. Link 2: BSL, with wet path length $L_{2}$, attenuation $A_{2}$ and elevation angle $\theta_{2}$.

**Figure 2 sensors-22-07019-f002:**
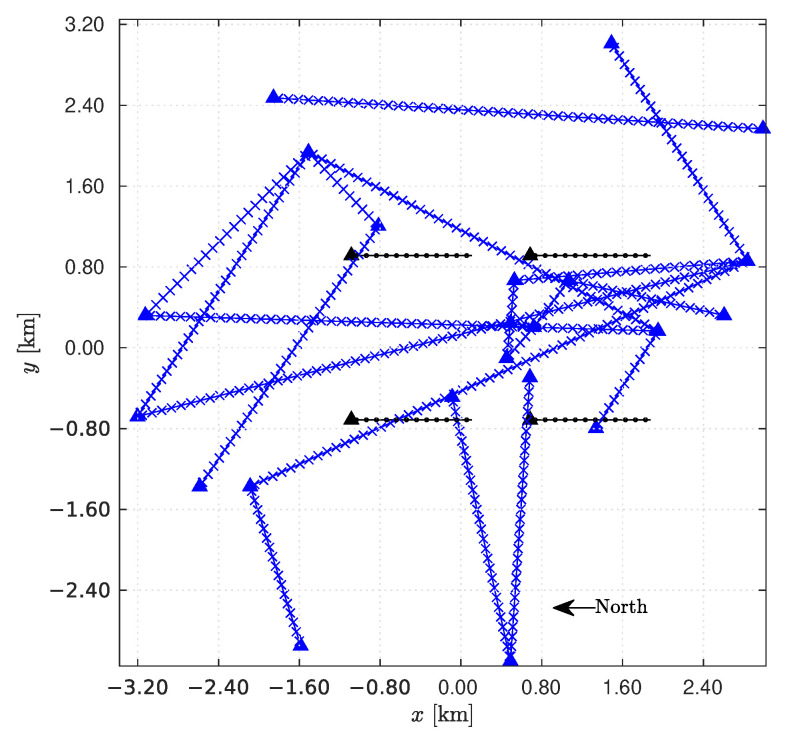
Network topology used for numerical performance evaluation via computer simulation. Legend: blue lines: CML links; black lines: ground projection of BSLs’ wet segments; ‘x’ marks: data points taken by the algorithm on CML links; ‘
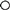
’ marks: ground projection of data points taken by the algorithm on BSLs’ wet segments; ‘
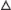
’ marks: receivers. Notice: CMLs are full duplex links with receivers on both endpoints of each link; BSLs are one-way links with receivers only on the ground endpoint of the link.

**Figure 3 sensors-22-07019-f003:**
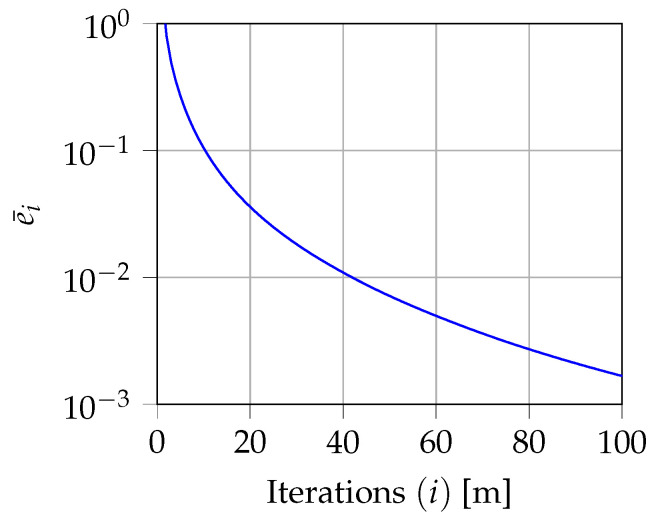
Mean error versus number of iterations.

**Figure 4 sensors-22-07019-f004:**
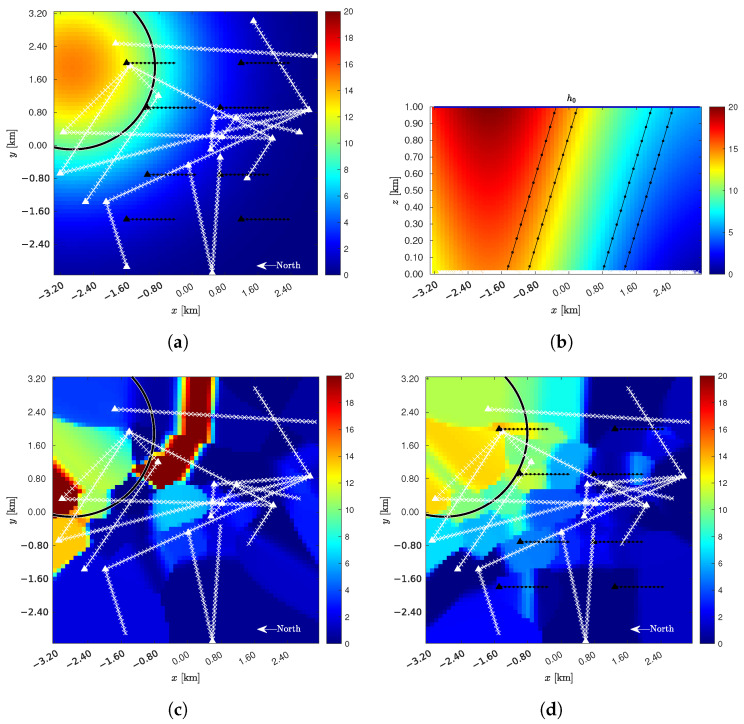
Rainfall rate estimation for a single realization with R=15 mm/h and $g_{\mathrm{alg}}=g_{G}=5$ mm/h/km. (**a**) $\bar{r}\left(C\right)$: $\pi_{H}\left(0\right)$ plane. (**b**) $\bar{r}\left(C\right)$: $\pi_{V}\left(y_{G}\right)$ plane. (**c**) r(C): $N_{\mathrm{CML}}=21$, $N_{\mathrm{BSL}}=0$, $\varepsilon_{\mathrm{RMS}}=5.715$, ρ=0.470. (**d**) r(C): $N_{\mathrm{CML}}=21$, $N_{\mathrm{BSL}}=8$, $\varepsilon_{\mathrm{RMS}}=1.981$, ρ=0.934.

**Figure 5 sensors-22-07019-f005:**
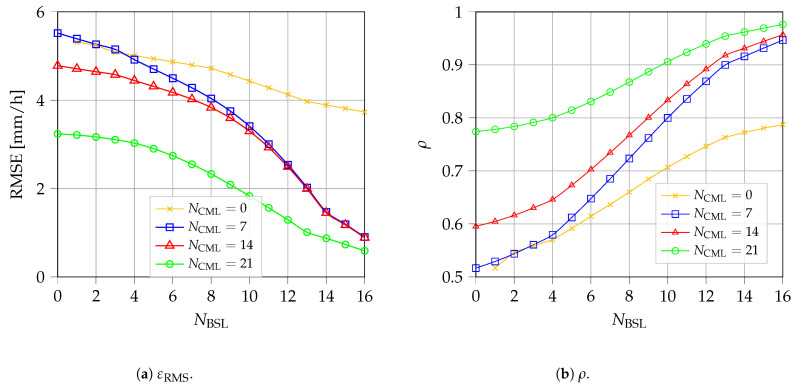
Estimation performance of the rainfall rate map vs. the number of BSL.

**Figure 6 sensors-22-07019-f006:**
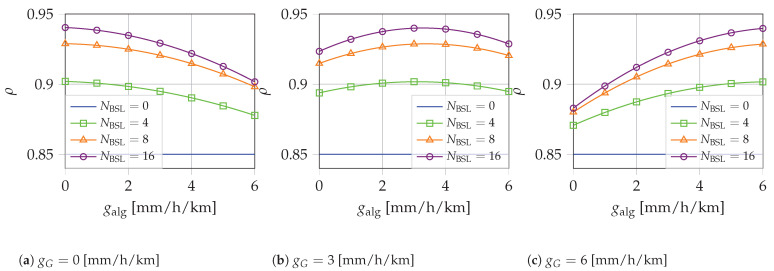
ρ vs. $g_{\mathrm{alg}}$ [mm/h/km], $N_{\mathrm{CML}}=21$, R=10 mm/h.

**Table 1 sensors-22-07019-t001:** Simulation parameters employed for every scenario.

*B*	40.96 km${}^{2}$	*D*	0.1 km	$h_{0}$	1 km
*J*	64	$f_{n}$	18 GHz	polariz.	vertical
$a_{n}$	0.0601	$b_{n}$	1.1154	$\sigma_{G}$	2 km

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
