# Peer review of "Rainfall Map from Attenuation Data Fusion of Satellite Broadcast and Commercial Microwave Links"

_sensors, 2022, doi:10.3390/s22187019_

Round 1

Reviewer 1 Report

This study presents novel algorithm for estimation of rainfall map based on BSLs and CMLs. The GMZ algorithm has been extended.

The manuscript can draw attention of researchers who are interested in the estimation of rainfall map. I recommend a major revision. The authors should address the following comments and revise their manuscript accordingly.

1.      It is suggested to carefully check whether all the notations are described in the text.

2.      Line 120: It seems that the discussions of numerical studies are too simple. The results in this study might be compared with those in the literature.

3.      Line 148: It is suggested to describe how to determine the simulation parameters in detail.

4.      It seems that the conclusions are too simple and need to be improved.

Reviewer 2 Report

The paper about Rainfall map from attenuation data fusion of Satellite Broadcast and Commercial Microwave links has an interesting topic and improve algorithm to estimate the rainfall rate map. the authors has structured well the entire paper but before be accepted to be published it is necessary to make some modifications:

1. The abstract of the paper needs important improvement. In this stage is just a description of the methods used without any connection with the results obtained.

2. the results are clearly expressed but the discussion part is missing. IN this part the authors must connect their results with other researches made in the field and also expressed the limitation of the methodology used.

3. How the authors verified their results? Could they explain this in the methodology part.

4. in the conclusion part the authors must express the practical application of the results obtained.

Round 2

Reviewer 1 Report

The authors have generally addressed the comments.